# Efficient Intrusion Detection System in the Cloud Using Fusion Feature Selection Approaches and an Ensemble Classifier

Mhamad Bakro [1,*], Rakesh Ranjan Kumar [1], Amerah A. Alabrah [2], Zubair Ashraf [3], Sukant K. Bisoy [1], Nikhat Parveen [4], Souheil Khawatmi [1] and Ahmed Abdelsalam [5]

1   Department of Computer Science and Engineering, Faculty of Engineering, C. V. Raman Global University, Bhubaneswar 752054, India; rakeshranjan.cdac@gmail.com (R.R.K.); sukantabisoyi@yahoo.com (S.K.B.); souheil.khawatmi@gmail.com (S.K.)
2   Department of Information Systems, College of Computer and Information Sciences, King Saud University, Riyadh 11543, Saudi Arabia; aalobrah@ksu.edu.sa
3   Department of Computer Engineering and Applications, GLA University, Mathura 281406, India; ashrafzubair786@gmail.com
4   Department of Computer Science and Engineering, Koneru Lakshmaiah Education Foundation, Guntur 522302, India; nikhat0891@gmail.com
5   Department of Software Engineering, LUT University, 53850 Lappeenranta, Finland; ahmed.abdelsalam@lut.fi
*   Correspondence: mhwb14794@gmail.com

**Abstract:** The application of cloud computing has increased tremendously in both public and private organizations. However, attacks on cloud computing pose a serious threat to confidentiality and data integrity. Therefore, there is a need for a proper mechanism for detecting cloud intrusions. In this paper, we have proposed a cloud intrusion detection system (IDS) that is focused on boosting the classification accuracy by improving feature selection and weighing the ensemble model with the crow search algorithm (CSA). The feature selection is handled by combining both filter and automated models to obtain improved feature sets. The ensemble classifier is made up of machine and deep learning models such as long short-term memory (LSTM), support vector machine (SVM), XGBoost, and a fast learning network (FLN). The proposed ensemble model's weights are generated with the CSA to obtain better prediction results. Experiments are executed on the NSL-KDD, Kyoto, and CSE-CIC-IDS-2018 datasets. The simulation shows that the suggested system attained more satisfactory results in terms of accuracy, recall, precision, and F-measure than conventional approaches. The detection rate and false alarm rate (FAR) of different attack types was more efficient for each dataset. The classifiers' performances were also compared individually to the ensemble model in terms of the false positive rate (FPR) and false negative rate (FNR) to demonstrate the ensemble model's robustness.

**Keywords:** cloud intrusion; ensemble learning; feature selection; weighted voting; crow search algorithm





## 1. Introduction

Cloud computing is a technology that provides internet services on demand with paid resources based on usage [1,2]. As cloud computing does not require large investments in infrastructure and development, government and private organizations have started to utilize the advantages of the cloud by deploying it in their operations [3,4]. The security of cloud computing is of prime importance, since individuals' and entities' information is stored in cloud data centers, which perhaps face security threats if a network is infiltrated by an attacker [5]. Since the network acts as the backbone of the cloud and facilitates the provision of cloud services to clients, any threats or weaknesses in the network directly affect the overall security and success of the cloud. Thus, safeguarding the network against any potential threats is a significant matter. The cloud utilizes an array of cybersecurity methods, including firewalls, Intrusion Prevention Systems (IPSs), and Intrusion Detection

Systems (IDSs), to tackle various security problems. Some of the attacks faced by cloud computing are denial-of-service (DoS) attacks, routing information attacks, distributed denial-of-service (DDoS) attacks, SQL injection attacks, cross-site scripting XSS attacks, etc. [6,7]. The foremost security challenge in cloud computing involves the detection and thwarting of network attacks. Lately, network intrusions have escalated due to insufficient countermeasures, and an IDS can address these security concerns. However, before deploying any IDS model in a cloud environment, it is crucial to ensure that the proposed system is well developed and performs effectively. Consequently, this paper aims to create such an efficient IDS.

An IDS is used to detect an intrusion in a network, monitor all packets, and decide whether any of the incoming and outgoing packets have been affected by the intrusion [8]. In conventional intrusion detection models, statistical and knowledge-based approaches were employed. These approaches struggled with the detection of unknown attacks and also faced difficulties processing large amounts of network traffic data [9]. Machine learning offers significant potential as a technique capable of developing a range of robust strategies to reinforce the security of systems such as cloud computing, IoT, and any other network-based system. A notable application of machine learning is in the design of an IDS. The role of this system is to analyze network traffic, distinguish between normal and abnormal behaviors, and appropriately categorize them [10]. ML approaches are considered to be better than conventional models since ML models possess the capability to learn complex traffic patterns and can precisely identify the attack [11]. Basic ML models face limitations when intrusion is more complex and diverse [8]. Studies conducted on conventional approaches show that ensemble models provide better results than single ML-based classifiers. Ensemble-based models are generated from numerous ML and DL models, so they offer a higher accuracy and a reduced false alarm rate (FAR) [12]. The ensemble-based models possess higher data processing capacities and better classification rates. Ensemble-based cloud intrusion detection has improved accuracy with a feature set formed by combining both the filter-based and automated feature selection models. Stacked autoencoder (SAE)-based automated feature selection helps to minimize the dimensions of the feature set by eliminating redundant and irrelevant features [13]. Conventional approaches have not focused on generating a comprehensive feature set. Since the selection of a better feature set ensures better classification results, both the filter and automated features are combined to make a set of features in this research. Our ensemble model employs classifiers such as SVM, LSTM, XGBoost, and FLN. The weights of the classifiers are optimized with CSA to enhance the classification results. In short, our main contributions are as follows:

1. An ensemble model is developed with the ML and DL models with SVM, LSTM, XGBoost, and FLN.
2. A comprehensive feature set is generated with the help of a filter and automated feature selection.
3. The weights of the selected ML and DL models are selected with the CSA.
4. We overcome the issue of unbalanced datasets by utilizing the SMOTE algorithm (Synthetic Minority Oversampling Technique), with the purpose of enhancing the detection accuracy for minority-type attacks.
5. The experiment is validated using old and modern benchmark datasets (NSL-KDD, Kyoto, and CSE-CIC-IDS-2018), that reflect a real-world environment involving the latest attacks. The outcomes are calculated in terms of accuracy, recall, false positive rate, false negative rate, precision, and F-measure.

We organize the rest of our work as follows: Section 2 details the related work, Section 3 explains the proposed model, Section 4 illustrates the experimental results, and Section 5 ends with the conclusions and future work.

## 2. Related Work

Ensemble techniques are built on the notion of combining several classifiers to achieve a competitive edge. Each classifier comes with its own set of benefits and drawbacks. Some

modules may be good at detecting certain types of attacks but not so good at detecting others. The ensemble approach picks the best result using a voting mechanism. Various studies have illustrated that ensemble approaches generally give better, more accurate outcomes than a single model [14]. Kushwah and Ranga [15] presented a voting extreme learning machine (V-ELM) for detecting DDOS attacks in cloud computing, which uses a large number of ELMs instead of a single ELM. Majority voting combines the results of all ELMs to obtain the final results. With multiple ELMs, the accuracy of detection increases and false alarms decrease. Srilatha and Shyam [8] proposed a cloud intrusion detection technique by combining kernel fuzzy c-means clustering (KFCM) and an optimal type 2 fuzzy neural network (O2TFNN). Lion optimization was employed to select the optimal parameters of the T2FNN. The performance of the method was simulated with the NSL-KDD dataset based on precision, recall, and F-measure. Xu et al. [9] suggested an intrusion detection model with a deep neural network (DNN). The improved LSTM model used in the proposed approach consists of gated recurrent units (GRU) along with a multilayer perceptron. The proposed LSTM-based approach achieved a higher detection rate for KDD 99 and NSL-KDD datasets. Mighan and Kahani [11] offered a deep-learning-based intrusion detection technique. A stacked autoencoder extracts the features, and classification approaches such as SVM, random forest (RF), decision forests, and naive Bayes were employed to obtain the classification results. The UNB ISCX 2012 dataset was evaluated with the proposed approach regarding accuracy, precision, sensitivity, and F-measure, in which it attained a better accuracy and reduced time consumption. Mayuranathan et al. [12] proposed an intrusion detection system using a restricted Boltzmann machine (RBM) model. The optimal features are selected with the random harmony search model, and the detection rate is enhanced with the seven extra layers in the RBM. In addition, the hyper-parameters are optimized for better results. Gaussian distribution replaces probability distribution in the proposed RBM model. Krishnaveni et al. [16] introduced an ensemble model for network intrusion with an efficient feature selection approach, in which the feature sets are reduced using univariate feature selection and classifiers such as SVM, naive Bayes, logistic regression, and decision trees are used as an ensemble model, with the class determined by a majority vote. The performance of this methodology, UEFFS, was evaluated with the NSL-KDD, Kyoto 2006, and real-time honeypot datasets, recording a high accuracy with a lower false alarm rate.

Thaseen et al. [17] demonstrated an ensemble of LSTM with a genetic algorithm (GA), in which the GA selects the features effectively. The ensemble of LSTM employs a voting approach based on the average probability combination rule. The intrusion detection technique evaluates the proposed model with criterion datasets, namely NSL-KDD and UNSW-NB15, in which a higher accuracy and detection rate and a minimum false alarm rate were achieved. The authors of [18] proposed an ensemble model for intrusion detection where the ensemble model is robust and able to produce results with less computation overhead. Parul Singh and Virendra Ranga [19] illustrated an ensemble learning approach in cloud computing, which used four machine learning techniques, namely boosted tree, bagged tree, subspace discriminant, and RUS boosted. The performance of the proposed approach was simulated with the CICIDS 2017 dataset and exhibited better results based on the detection rate and execution time. Mehanovic et al. [20] displayed an efficient feature selection for intrusion detection. The features were reduced with ML approaches such as SVM, artificial neural networks (ANNs), random tree (RT), logistic regression, and naive Bayes. The parallel genetic algorithm was implemented with an open-source MapReduce library that helps select better features. The authors of [21] presented a network intrusion detection mechanism where a CNN model based on contextual feature extraction improves the accuracy of the IDS. The approach reduced the feature set before feeding it into a CNN to reduce the computational time. Tummalapalli and Chakravarthy [22] exhibited Bayesian fuzzy clustering and a two-level gravitational group search-based support vector neural network (GG-SVNN). In this approach, intrusion information is gathered with a level

1 classifier, and level 2 classifier makes the final decision on the presence or absence of intruders. The proposed approach achieved a better accuracy and a lower false alarm rate.

Amali Angel Punitha and Indumathi [23] proposed ensemble-based attack detection. The approach consists of centralized cloud information and accountability integrity with an imperialist competitive key generation algorithm (CCIAI-ICKGA) proposed for key generation. The attack detection rate is high and the computation time is less than the current methods. Su et al. [24] employed a DL model to detect intrusion in the cloud network, which uses the BAT model as a combination of bidirectional LSTM and an attention mechanism. The local features of the traffic data are captured by the multiple convolutional layers. Softmax classifies the network traffic. The experimental results proved that the approach achieved a higher accuracy. Bhati et al. [25] implemented an ensemble-based intrusion detection approach with XGBoost. The ensemble-based XGBoost focused on improving the selection of features and the accuracy. The bias–variance tradeoff is smoothened with the tree-boosting machine learning algorithm. The approach was implemented on the KDDcup99 dataset and achieved an accuracy of 99.95%. Suman et al. [26] presented a multi-objective optimization (MOO) to select features. The optimal feature sets were obtained with the nondominated sorting genetic algorithm (NSGA-II). The optimal feature sets were applied to ML classifiers such as decision tree, SVM, RF, k-nearest neighbor (KNN), Adaboost, etc., to determine the effect of optimized features on various ML classifiers. The approach was simulated with various datasets such as Kyoto 2006+, NSL-KDD, and KDD-99. Rajagopal et al. [27] recommended a stacking-ensemble-based paradigm for intrusion detection systems (IDS). They stressed that this limitation is a significant component in a stacked ensemble's performance when compared to the single best learner. Lopez et al. [28] introduced an NIDS utilizing a conditional variational autoencoder (CVAE). Their proposal marked the first application of a CVAE, which offered the ability to retrieve missing categorical features from incomplete training datasets. The proposed solution features a unique architecture that embeds intrusion labels into the decoder layers. When tested on the NSL-KDD dataset, the model outperformed other well-known classifiers, achieving an accuracy rate of up to 99%.

Zhou et al. [29] presented an IDS approach that leverages feature selection and ensemble learning methods. Initially, they proposed a correlation-based feature selection combined with a bat algorithm (CFS-BA), aimed at identifying the ideal subset. Subsequently, they designed an ensemble classifier using C4.5, random forest (RF), and forest by penalizing attributes (Forest PA), along with the AOP rule to construct the classification model. Ultimately, they applied a voting strategy to merge the probability distributions of foundational learners for detecting attacks. The validation was conducted using the NSL-KDD, AWID, and CIC-IDS2017 datasets. Although the overall performance was superior, the accuracy of the proposal for minority classes (R2L and U2R) in the NSL-KDD dataset was not satisfactory due to inadequate addressing of the imbalanced data issue. Balyan et al. [30] presented an effective hybrid network-based IDS (HNIDS) model, created using a combination of the enhanced genetic algorithm, particle swarm optimization (EGA-PSO), and the improved random forest (IRF) method. Initially, the HNIDS applied the hybrid EGA-PSO method to bolster minor data instances, thereby creating a balanced dataset. The PSO method assists in vector enhancement, while the GA is improved with a multi-objective function for optimal feature selection. Subsequently, the IRF method is employed as a classifier. The model's efficiency was evaluated using the NSL-KDD dataset in both binary and multi-class classification scenarios. Bakro et al. [31] suggested an IDS that utilizes filter methods (information gain, gain ratio, and chi-square) to select relevant features and employs the SVM model for classification. However, they did not tackle the issue of unbalanced data. Their model underwent testing using the NSL-KDD, Kyoto, and CSE-CIC-IDS-2018 datasets.

From the preceding relevant works and methodologies outlined in Tables 7–9, we noted some constraints, such as the inadequate handling of unbalanced data issues and the use of a single classifier instead of an ensemble approach, which may not yield satisfac-

tory performance across all categories of the used dataset. Moreover, we have discerned that designing an efficient intrusion detection system necessitates well-prepared datasets, and issues related to imbalanced data must be adequately addressed. Additionally, the reduction of high-dimensional datasets through feature selection methods also plays a pivotal role in the success of the IDS model; therefore, we found that the majority of these studies employed various techniques to select optimal features, including wrapper, filter, metaheuristic, and unsupervised models, among others. As a result, in our research, we constructed our proposal by leveraging the strengths and addressing the potential weaknesses of other studies to achieve superior results.

## 3. Methodology

The proposed cloud intrusion detection system consists of pre-processing to reduce the class imbalance problem. The final feature set is formed with the combination of filter and automatic feature selection approaches.

Figure 1 presents a model of the proposed technique. The ML and DL models are trained with the feature set. Then, the weights of the ML and DL models are optimized with the CSA approach, and the final classification results are obtained from the ensemble model using the weighted voting concept.

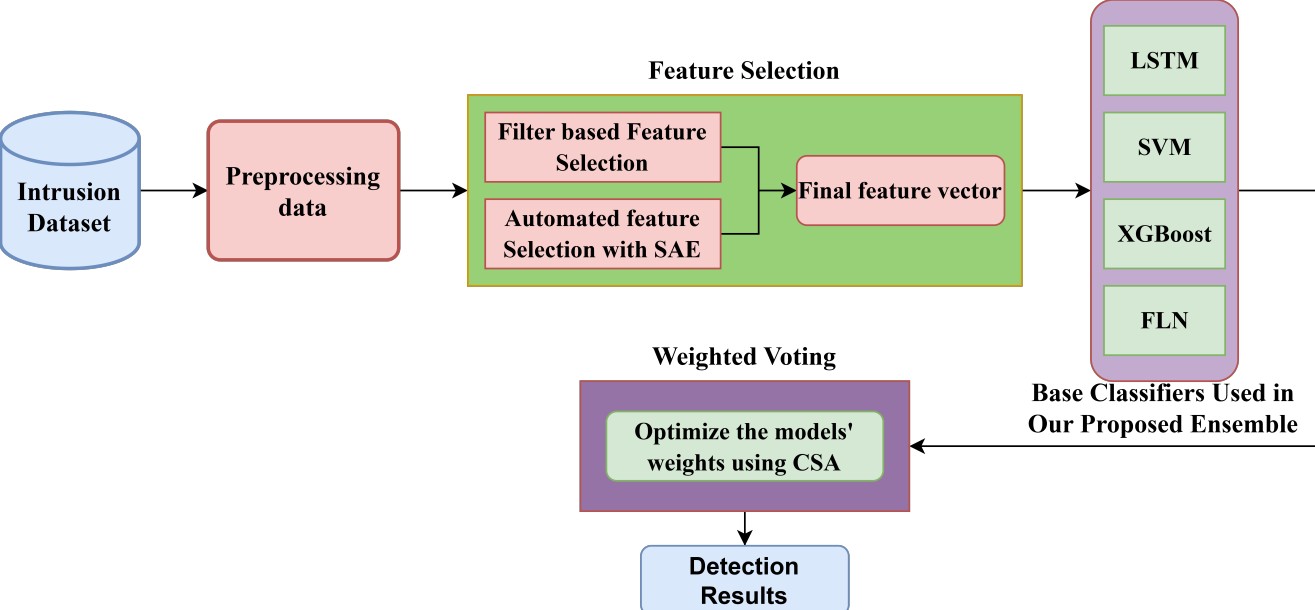

**Figure 1.** Proposed Block Diagram.

### 3.1. Pre-Processing

This section details the pre-processing of the data in order to prepare them for feature selection. The creation of a better training dataset is also required [32]. The dataset contains a variety of symbolic, binary, numerical data, etc. Some of them may have imperfect and inconsistent values that lead to unsatisfactory outcomes. The proposed system only takes clean and numeric inputs; thus, the dataset must be prepared first. Therefore, we implemented the following steps:

- Removing the noisy, incomplete outliers and irrelevant, duplicate values and deleting features: In NSL-KDD we deleted features such as "difficulty_level", and in the Kyoto dataset, we omitted features such as "Source IP Address", "Destination IP Address", etc. Furthermore, "Timestamp", "Destination Address", "Source Address", "Source Port", and other features were removed in CSE-CIC-IDS 2018 [33], that means we only retain one useful value.
- Feature encoding, also known as feature mapping, which means transforming categorical variables into numerical form. There are two methods used for this, one-hot

encoding and ordinal encoding, in which one-hot encoding yields a higher classifier performance than ordinal coding [34]. As a result, one-hot encoding was adopted in this study. Protocol type, service, and flag are three symbolical characteristics in the NSL-KDD dataset. For example, the feature of protocol type has three category values (TCP, UDP, and ICMP). They form binary vectors (1, 0, 0), (0, 1, 0), and (0, 0, 1) after one-hot encoding [35], and we will perform the same operation on the Kyoto dataset. In the CSE-CIC-IDS 2018 dataset, we implement one-hot encoding on a protocol, that is a nominal feature that has three groups (protocol 0.0: hop-by-hop IPv6 "HOPOPT", protocol 6.0: TCP, and protocol 17.0: UDP) [36].

- Feature scaling, also known as data normalization, which is a technique to convert the whole scope of values from a set of features into a predetermined range. Common methods of feature scaling are normalization and standardization. Standardization is called also Z-score normalization, where a single standard deviation and values centered around the mean indicate that the attribute's mean tends to zero, and the resulting distribution has a standard deviation of one unit. Normalization or Min-Max scaling is the process of shifting and rescaling values so that they fall in the range [0,1], which offers satisfying outcomes in the AE procedure [36]. In our research, we used Min-Max scaling, shown in the following equation:

$$X_j^* = (X_j - X_{\min}) / (X_{\max} - X_{\min}) \tag{1}$$

where $X_j$ is the initial value, $X_j^*$ are the data after the process, $X_{\min}$ is the impacting data sequence's lowest value, and $X_{\max}$ is said sequence's highest value [37].

- Depending on the attack type, the label/class field must be gathered. The class column is split into two groups for detection: normal and abnormality. Regarding attack classification, the class column is separated into types as follows: 4 types (DOS, Probe, R2L, and U2R) in NSL-KDD [38], 2 types (known and unknown attacks) in the Kyoto dataset, and 14 types (DDOS attacks-LOIC-HTTP, DDOS attack-HOIC, DDOS attack-LOIC-UDP, FTP-Brute Force, SSH-Brute force, Brute Force-XSS, Brute Force-Web, DOS attacks-SlowHTTPTest, DOS attacks-Hulk, DOS attacks-GoldenEye, DOS attacks-Slowloris, Infilteration, Bot, and SQL Injection) in the CSE-CIC-IDS 2018 dataset.

- Feature correlation. This is a useful technique for feature engineering, and is a statistical method that defines the relevance between one or more variables in order to detect the related important features and only keep these features [39].

After that, the pre-processed data are split into two sets: a training dataset and a test dataset. In order to address the class imbalance challenge which affects the performance of the classifiers for a minority of attacks [40], we employ the Synthetic Minority Over-Sampling Technique (SMOTE) by expanding the training dataset's mitotic instances [41]. In the following step, we extract the features from the training dataset that will be used to train the proposed model. The test dataset, which generates various classifications, is then used to test the trained proposal.

*3.2. Feature Selection*

In the proposed method, after pre-processing, feature selection is handled. Feature selection (FS) is considered a pre-processing approach that aids in the selection of suitable features. Feature selection methods minimize data dimensionality by removing redundant and irrelevant attributes [13]. When irrelevant features are added to the classification process, not only is the accuracy of the process reduced, but it also takes up more space and time to execute. As a result, feature selection is critical since it allows for a deeper understanding of data by holding only the relevant appropriate features, therefore improving the classifier's accuracy, speed, and predictive capacity [42]. One feature selection approach is the filter technique, which shortlists the essential attributes first, irrespective of any classification method. Variable ranking approaches are used in filter methods to score

the variables. The most important variables are selected from a list of variables, leading to the elimination of less relevant attributes [43]. Therefore, the final feature vector is obtained from the filter and automated feature selection.

### 3.2.1. Filter Methods

Filter-based feature selection methods such as information gain (IG), gain ratio, chi-squared, and symmetric uncertainty are used to select the features, and are given as follows:

Information Gain

Information gain (IG) is a well-known filter-based feature selection approach. IG can reduce the noise caused due to irrelevant features. The execution time and computational complexity of IG are low. Features with a high IG are relevant. IG calculates the entropy change after utilizing the attribute, and thus it illustrates the importance of a specific feature. The impurity of a feature is measured in entropy. The higher the entropy, the more information the feature has; therefore, the entropy determines the best feature, which means that it will help predict the class label [26]. The entropy of a sample S is calculated according to the following:

$$Entropy(S) = -\sum_{i}^{n} P_i \log_2 P_i \tag{2}$$

where $n$ and $P$ refer to the number of classification values and the number of samples in class $i$, respectively. The information gain is given as:

$$IG(S, A) = Entropy(S) - \sum Values(A)\left(\frac{|S_n|}{|S|}\right) Entropy S_n \tag{3}$$

where $A$ is the attribute, $Values(A)$ is the attribute's potential value, $|S_n|$ is the number of samples for the value $n$, and $|S|$ is the total number of samples. $Entropy S_n$ is the entropy for a sample that has a value of $n$. According to [42], the IG of feature $A$ is given as [44]:

$$IG(A) = Info(M) - Info_A(M) \tag{4}$$

where the entropy of the absolute dataset is $Info(M)$ and the entropy of attribute $A$ is $Info_A(M)$.

Gain Ratio

Gain ratio (GR)-based feature selection improves the IG method. The GR is high when the data are evenly spread and it takes a small value when data come from only one branch of the attribute [13]. The GR is determined with the help of split information [16].

$$Split\ info_A(M) = -\sum \frac{|M_n|}{|M|} * \log_2 \frac{M_n}{M} \tag{5}$$

where $|M|$ is the number of probable values $x$ can select and $|M_n|$ is the number of actual values of feature $x$. The gain ratio is obtained as:

$$Gain\ ratio(A) = IG(A)/Split\ Info(A) \tag{6}$$

Chi-Squared

The chi-square method is one of the feature selection approaches that is used to find the relationship between two variables. The independence between two events can be

found using this approach, and it may help to determine a feature's independence from its class [16]. The chi-squared is given as:

$$\gamma^2 = \sum_{iz} (O_{iz} - E_{iz})^2 / E_{iz} \qquad (7)$$

where $i$ and $z$ represent two variables, $O$ and $E$ are the observed and expected value, respectively, and $\gamma^2$ indicates the Chi-squared value.

Symmetric Uncertainty

This form of feature selection is employed to evaluate the ranking of the outcome. The symmetric uncertainty is given as [16]:

$$SU(i,j) = 2 * IG(i/j)/H(i) + H(j) \qquad (8)$$

where $H(i)$ and $H(j)$ represent the entropy of features $i$ and $j$, respectively.

3.2.2. Automated Feature Selection with Stacked Autoencoder

An autoencoder is a common unsupervised DL approach that belongs to the family of artificial neural networks with three layers: input, hidden, and output layers [11]. The objective of the autoencoder is to obtain a similar reconstructed output and input, thereby attaining feature selection and a dimensionality reduction of the input. Autoencoders are able to learn efficient and compressed representations for the collection of data. In an AE, the data are reconstructed after computations based on the concept of learning the best features and matching the output to the input as closely as feasible [45]. Stacked AEs are one of the AE types designed for automated feature selection [46]. In an SAE, the output of the first AE is the input of the second one. This is repeated based on the depth of the architecture. The original information is encoded by the encoder to acquire high-level features in the middle layer, and the decoder reconstructs the input information. The hidden layer for the original training data x is given as follows [47]:

$$h^{(i)} = f\left(W_1^T x^{(i)} + b_1\right) \qquad (9)$$

The activation function is represented by $f = \tanh(\circ)$. The output obtained with the decoding function is calculated as:

$$z^{(i)} = W_2^T h^{(i)} + b_2 \approx x^{(i)} \qquad (10)$$

where $W_1^T$ and $W_2^T$ are the weight matrices and $b_1$ and $b_2$ are the different bias vectors. In order to train the AE, the expression is minimized as follows:

$$L(X,Z) = 1 \Big/ 2 \sum_{i=1}^{n} \left\| x^{(i)} - z^{(i)} \right\|^2 \qquad (11)$$

where $x^{(i)}$ and $z^{(i)}$ denote the $i$th element of $x$ and $z$, respectively, and $n$ signifies the input set denoted by $x^{(i)}$ where $i = 1 \ldots n$.

Prior information is ignored in the AE, while an SAE learns a more satisfactory representation of the input data than a single AE. The hidden neurons in the proposed SAE for the two encoder layers are 50 and 30, respectively, as shown in Figure 2. An SAE functions essentially as a multi-layer AE. In this setup, the output from one layer provides the input for the subsequent layer. Typically, an SAE undergoes layer-by-layer training, a procedure commonly known as pretraining. Following this stage, the pre-trained network weights are amalgamated to establish the ultimate network weights. When using an SAE for feature selection, we begin by training the SAE on the input data, after which we utilize the encoded representations produced by the SAE as the new feature set. This new feature set possesses a smaller dimensionality and includes the most crucial attributes of the input

data. Consequently, this can enhance the efficiency and performance of the classification model. The filter and automated features of SAE are combined to form the feature vector.

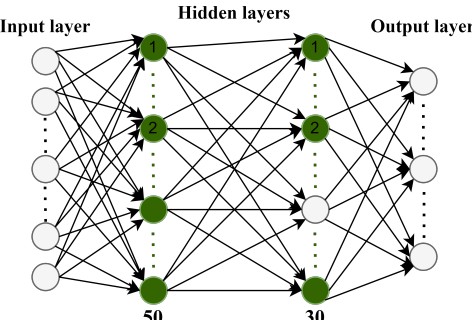

**Figure 2.** Stacked autoencoder.

*3.3. Classification Using the Ensemble Learning Approach*

After the feature selection process, we propose an ensemble learning approach that includes several base classifiers, namely LSTM, SVM, XGBoost, and FLN. Each of these classifiers is trained using optimally selected features to solve the same problem, and their outputs are then combined to enhance the overall results. Two commonly used methods for combining predictions from different models are majority voting and weighted voting. Majority voting is ideal when all models in the ensemble have equal reliability. Conversely, if certain models consistently outperform others, weighted voting becomes the superior choice. In our ensemble learning approach, we have chosen to employ weighted voting due to the different performance levels of our base classifiers. Before detailing our chosen weighted voting method in Section 3.4, we provide a brief explanation of each model utilized in our proposed ensemble below.

3.3.1. LSTM

Long short-term memory is a form of recurrent neural network (RNN) that belongs to the family of supervised DL [48]. Compared to other RNNs, LSTM offers more advantages [33]. LSTM was developed to tackle the vanishing gradient problem that is encountered in RNNs through a concept known as gates. These gates perform an important role in prediction because they learn to keep relevant data while discarding irrelevant inputs [17]. LSTM consists of input, hidden, and output layers. One cell state layer and three gate layers make up the hidden layer: an update cell state, a forget gate layer, an input gate layer, and an output gate layer [49]. The mathematical model of the cell state layer and the three gate layers is given as [50]:

$$
\begin{aligned}
&\textit{Input gate}: i_t = \sigma\left(W^{(i)}X_t + U^{(i)}S_{t-1}\right) \\
&\textit{Forget gate}: f_t = \sigma\left(W^{(f)}X_t + U^{(f)}S_{t-1}\right) \\
&\textit{Output gate}: o_t = \sigma\left(W^{(o)}X_t + U^{(o)}S_{t-1}\right) \\
&\textit{The memory cell's new state}: \widetilde{S}_t = \tanh\left(W^{(c)}X_t + U^{(c)}S_{t-1}\right) \\
&\textit{The memory cell's final state}: S_t = f_t \circ S_{t-1} + i_t \circ \widetilde{S}_t \\
&\textit{The memory unit's final output}: O_t = Soft\max(o_t \circ \tanh(S_t))
\end{aligned}
\tag{12}
$$

where $W^i$, $W^f$, $W^o$, $W^c$, $U^i$, $U^f$, $U^o$, and $U^c$ are weight matrices, $X_t$ is the input vector, "$\circ$" represents the Hadamard product, and "$\sigma$" denotes Sigmoid. In the final output layer, we utilize Softmax for the classification [9].

3.3.2. SVM

Support vector machine is a supervised ML technique utilized for classification and regression. The search for a max-margin separation hyperplane in the n-dimension space

of features is the used approach in SVMs; therefore, we can obtain satisfying outcomes regardless of the size of the dataset [51]. Using support vectors, the hyperplane acts as a decision limit [20]. Both linear and nonlinear issues are addressed using this method. Kernel functions are applied in nonlinear situations; kernel techniques are commonly utilized in SVMs and other machine learning models [13]. SVMs were the most effective among ML algorithms in past years, until the appearance of DL [33]. SVMs are also employed in multi-class problems. The hyperplane is given as [52]:

$$H(x) = w^T(x) + b \tag{13}$$

where $x$, $w$, and $b$ denote the input, weight, and bias values, respectively. The objective of SVM is calculated as follows [53]:

$$Minimize : \frac{1}{2}\|w\| + C\left(\sum_i \mu_i\right) \tag{14}$$

$$Subject\ to : z_i\left(w^T x_i + w_o\right) \geq 1 \tag{15}$$

where $\mu_i$ is the training error, $z_i$ is the class label, $x_i$ is the feature vector, $i$ is the sample number, and $C$ is the cost parameter.

### 3.3.3. XGBoost

Extreme gradient boosting utilizes a tree ensemble model, which means XGBoost is a version of a multiple decision tree, because one tree may not be sufficient to provide satisfactory outcomes [54]. Decision trees belong to the class of supervised ML methods that are employed in regression and classification. To overcome the over-fitting issue, the DT algorithm automatically chooses the optimal attributes and removes irrelevant data to create a decision tree [45]. XGBoost can reduce the processing time and also possesses a higher accuracy. The accuracy is enhanced because XGBoost uses a second-order Taylor expansion [33]. The calculation is parallelized and the speed is increased as the feature is sorted and blocked by XGBoost. The output is predicted by XGBoost for a dataset that has k features. The $N$ additive function to predict the output is given as follows [25]:

$$\widehat{y}_i = \phi(x_i) = \sum_{n=1}^{N} f_n(x_i), f_n \in F \tag{16}$$

where $f$ represents the tree structure and $D$ is the ensemble model of the tree. The objective is minimized according to the following:

$$L^{(t)} = \sum_{i=1}^{n} \left(g_i f_t(x_i) + \frac{1}{2}h_i f_t^2(x_i)\right) + \Omega(f_t) \tag{17}$$

### 3.3.4. FLN

Fast-learning networks are considered a double parallel forward neural network (FNN) in which a direct connection is added along with the input and output layers. There are three layers to the FLN: input, hidden, and output layers. The nonlinear relationship from the hidden to the output layer is merged in an FLN, as is the linear relationship from the input to the output layer. FLNs are considered a development of extreme learning machine (ELM) algorithms, where ELM algorithms were proposed to avoid the drawbacks of ANNs which is one of the supervised ML models. FLNs demonstrate excellent performance while consuming less time [45]. The FLN model is given as follows [37]:

$$T_i = aX_i + \sum g\left(W_j X_i + b_j\right)\beta_j \ , \ i = 1, 2, \dots q \tag{18}$$

In an FLN model, the weights are randomly created along with the hidden layer thresholds. The hidden layer output matrix $H$ is calculated. The output weight matrix is finally formed, where we use SoftMax in the output layer with respect to the classification.

### 3.4. Crow Search Algorithm (CSA) and Weighted Voting

The objective of ensemble-based classification is to create a model with a high prediction capacity system. Each of the individual classifiers varies regarding the number of layers, pooling, filter size, etc. The predictions of individual DL and ML models are combined with the voting technique. In an ensemble model, majority voting and weighted voting are generally used. In majority voting, each classifier in the ensemble casts one vote. The class label that receives more than half of these votes, drawn from the set of predicted class labels from different classifiers, is then selected. In the weighted voting approach, base classifiers are assigned differing weights, reflecting their performance. Each classifier's vote is then multiplied by this weight. The class that accumulates the highest total weighted vote becomes the ensemble's final prediction. Therefore, the weighted voting assignment of weights on the classifiers has a considerable effect on the prediction of the ensemble model. The optimized weights are not randomly generated; the weights are obtained with an objective function so that weights will be tuned to improve the performance of the classifiers. Weight optimization is carried out with several optimization techniques such as genetic algorithms, particle swarm optimization, and differential evolution algorithms. Here, weighted voting with the CSA algorithm is proposed.

The prediction of individual models can be handled inside the ensemble. The ensemble takes into account all the classifiers' prediction probabilities and decides the final prediction result based on the majority of the prediction probability results. The weight-based voting approach is chosen to assign weights to the classifiers to select the final prediction. The weights are decided with a meta-heuristic algorithm: the crow search algorithm (CSA). The CSA works based on the foraging pattern of crows. The crows represent the solution in this optimization approach. The initial position is randomly assigned, and the fitness value is calculated for the memorized position [55]. In the CSA, based on two conditions, the positions of the crows are determined. In the first condition, a crow is unaware that it is being followed by another crow to the location of the hidden food. In the second condition, the crow possesses the knowledge that another crow is following it, and to avoid theft, the crow will move to some other position in the search space [56]. The position of the crows in the search space is given as follows:

$$P_i = (Wb_1, Wb_2, \ldots Wb_n) \tag{19}$$

where $P_i$ is the position of the $i$th crow and $Wb_n$ is the weight of the nth base classifier. The objective function of the weighted voting with the CSA is:

$$f(P_i) = 1 - \frac{\sum_{k=1}^{t} (P_n - a_n)^2}{s} \tag{20}$$

where $p_n$ is the predicted value of the class after the application of CSA-based weighted voting, $a_n$ is the actual value of the class, and $s$ is the number of test samples. Based on the objective function, the CSA updates the crow's position. The optimal weights are thus obtained from the CSA-based weighted voting approach for every classifier in the ensemble model. A final prediction is carried out with the optimized weights through weighted majority voting of the results obtained from the individual classifiers.

### 4. Experimental Results

In this part, we detail the datasets that we utilized, followed by the performance metrics. The outcomes of our proposal are then discussed, and finally, the constraints are shown. Our proposal was executed using libraries such as Scikit-learn and Tensorflow (ML-DL framework), and we used Imblearn to address the unbalanced dataset issue. The simulations were carried out in Python 3.9.0, with the Google Co-laboratory Pro platform with 25 GB RAM on an Intel Core i5 processor (ASUS, Bhubaneswar, India).

*4.1. Dataset Description*

4.1.1. NSL-KDD

This dataset is an upgraded version of KDD'99 that was developed by Tavallaee et al. [57] to handle some of the innate drawbacks of the KDD'99 dataset, where duplicate and redundant records have been removed in order to reduce the issue of data bias to make benchmarking learning algorithms more acceptable and realistic [58]. Thus, it is most commonly utilized in much research on intrusion detection [45]. We utilized KDDTrain+ and KDDTest+ as the training and test sets in the NSL-KDD dataset [59], which makes it useful to perform experiments. Every record is labeled as normal or abnormal based on the 42-feature set, and the abnormal type is divided into four kinds, namely DOS, probe attacks, remote to local (R2L) attacks, and user to root (U2R) attacks [57]. Although this dataset is perhaps not a perfect expression of real computer networks, we feel it is still a useful baseline for researchers to evaluate alternative methodologies [60]. Table 1 represents the NSL-KDD.

**Table 1.** Composition of the NSL-KDD dataset.

|  | **KDDTrain+** | **KDDTest+** |
|---|---|---|
| Normal | 67,343 | 9711 |
| DOS | 45,927 | 7460 |
| Probe | 11,656 | 2421 |
| R2L | 995 | 2885 |
| U2R | 52 | 67 |
| **Total** | **125,973** | **22,544** |

4.1.2. Kyoto

The Kyoto dataset was collected between 2006 and 2015 from a honeypot system [61], email servers, web crawlers, darknet sensors, and other servers installed at Kyoto University. In our research, we used the most recent dataset that contains traffic statistics through to 2015. The dataset has 24 features, 14 of which were gathered from KDD-99, and the other 10 are modern features that allow us to more properly investigate what occurred in our networks. Each record is classified according to a label, where 1 refers to normal traffic, $-1$ points out known attacks, and $-2$ indicates an unknown attack [26]. In the Kyoto dataset, there is no clear demarcation between the training and test sets and the dataset is extremely huge. As a result, we chose a random subset to test our assumptions, which is shown in Table 2.

**Table 2.** Composition of the Kyoto dataset.

|  | **Train** | **Test** |
|---|---|---|
| Normal | 45,260 | 11,250 |
| Known attack | 303,412 | 75,945 |
| Unknown attack | 11,095 | 2747 |
| **Total** | **359,767** | **89,942** |

4.1.3. CSE-CIC-IDS-2018

The Canadian Institute of Cybersecurity (CIC) with the Communications Security Establishment (CSE) created this dataset, which is a modernised version of CIC-IDS 2017 [62]. It is the most up-to-date launched dataset for intrusion detection research, which reflects the present real network environment, including seven traffic categories [36]: Benign, Botnet, Web Attacks, DDoS, DoS, Brute Force, Infilteration, and 14 types of labeled attacks with 84 features [63]. Additionally, there is no formal distinction between training and test sets in CSE-CIC-IDS 2018, and it is an excessively large dataset. Therefore, we randomly selected a subset to test our proposal, which is indicated in Table 3.

**Table 3.** Composition of the CSE-CIC-IDS 2018 dataset.

| Category | Attack Type | Training | Test |
|---|---|---|---|
| Benign | - | 144,198 | 36,003 |
| Botnet | Bot | 14,602 | 3759 |
| Web Attack | Brute Force-Web | 611 | 160 |
| | Brute Force-XSS | 230 | 51 |
| | SQL Injection | 87 | 24 |
| DDOS Attack | DDOS attack-HOIC | 34,376 | 8536 |
| | DDOS attack-LOIC-UDP | 1730 | 450 |
| | DDOS attacks-LOIC-HTTP | 28,906 | 7215 |
| DOS Attack | DoS attacks-GoldenEye | 8185 | 2059 |
| | DOS attacks-Hulk | 23,113 | 5796 |
| | DOS attacks-SlowHTTPTest | 7068 | 1753 |
| | DOS attacks-Slowloris | 4003 | 992 |
| Brute force | FTP-BruteForce | 9998 | 2494 |
| | SSH-Bruteforce | 9973 | 2439 |
| Infilteration | Infilteration | 8107 | 2066 |
| **Total** | **-** | **295,187** | **73,797** |

### 4.2. Performance Metrics

In this passage, we explain the evaluation measures for assessing the performance of our proposed model, which are calculated based on the various variables in the confusion matrix, a two-dimensional matrix that describes the actual/predicted classes, as is presented in Table 4 [45]. Correct identification of the data instances by classifiers as an attack is denoted a True Positive (TP) case, while if correctly identifying it as normal it is denoted a True Negative (TN). However, incorrect identification of the data instances by classifiers as an attack is denoted a False Positive (FP), while if correctly identifying as normal it is denoted a False Negative (FN) [31].

**Table 4.** Confusion matrix.

| | Predicted (Attack) | Predicted (Normal) |
|---|---|---|
| Actual **(Attack)** | True Positive (TP) | False Negative (FN) |
| Actual **(Normal)** | False Positive (FP) | True Negative (TN) |

- **Accuracy (ACC):** It is the proportion of instances that have been correctly detected to the overall number of instances. It is determined as follows:

$$ACC = \frac{TP + TN}{TP + TN + FP + FN} \tag{21}$$

- **Recall (R):** It is the proportion of successfully recognized normal instances to the total number. Furthermore, it is called the detection rate (DR), sensitivity (S), or true positive rate (TPR). It is calculated according to:

$$R = \frac{TP}{TP + FN} \tag{22}$$

- **False Alarm Rate (FAR):** It calculates the ratio of records that are mistakenly detected as attacks to all records. Furthermore, it is names the false positive rate (FPR). It is estimated as follows:

$$FAR = \frac{FP}{FP + TN} \tag{23}$$

- **False Negative Rate (FNR):** It is also known as the missed alarm rate (MAR), and is the proportion of instances that are wrongly detected as normal to all instances. It is given as follows:

$$FNR = \frac{FN}{FN + TP} \tag{24}$$

- **Precision (P):** It is the ratio of accurately predicted attacks to all attack samples, which provides how accurate predicted positive values are. It is evaluated as follows:

$$P = \frac{TP}{TP + FP} \tag{25}$$

- **F-measure (F):** It is utilized to assess the accuracy of a detection system taking into account precision and recall, also known as the F1 score. It is computed according to:

$$F - measure = \frac{2(precision * recall)}{(precision + recall)} \tag{26}$$

*4.3. Results Discussion*

By the simulation, the performance metrics of multi-class classification are calculated for our proposed model in terms of accuracy and the macro-average of recall, precision, and F1-score.

Table 5 presents the performance of the various feature selection approaches such as filter methods, stacked autoencoder, and fused features. It is observed that the fused feature method has succeeded in exhibiting better results than the others.

**Table 5.** Comparison of performance with filter-based feature selection, automated feature selection with SAE, and fused features.

| Method | Dataset | No. of Features | ACC | R | P | F |
|---|---|---|---|---|---|---|
| **Filter methods** | NSL-KDD | 50 | 97.99 | 90.42 | 96.44 | 93.13 |
| | Kyoto | 10 | 96.52 | 95.81 | 94.42 | 94.88 |
| | CSE-CIC-IDS-2018 | 6 | 98.90 | 92.75 | 99.50 | 95.17 |
| **Automated with SAE methods** | NSL-KDD | 93 | 98.41 | 94.33 | 98.33 | 96.17 |
| | Kyoto | 33 | 97.67 | 96.96 | 96.14 | 96.43 |
| | CSE-CIC-IDS-2018 | 6 | 99.02 | 97.86 | 99.91 | 98.79 |
| **Fused features** | NSL-KDD | 45 | 99.01 | 99.08 | 99.95 | 99.51 |
| | Kyoto | 15 | 98.99 | 98.93 | 98.16 | 98.53 |
| | CSE-CIC-IDS-2018 | 6 | 99.99 | 99.87 | 99.96 | 99.91 |

Table 6 presents the performance of the proposed method with and without the CSA. The proposal without the CSA used random weights, while with the CSA approach optimized weights have been generated. Therefore, CSA-based weighted voting showed a better performance than the method without the CSA.

**Table 6.** Comparison of performance when the CSA is not used and when the CSA is used.

| Method | Dataset | ACC | R | P | F |
|---|---|---|---|---|---|
| **Without CSA** | NSL-KDD | 97.01 | 90.31 | 95.45 | 92.61 |
| | Kyoto | 96.19 | 95.52 | 91.92 | 93.25 |
| | CSE-CIC-IDS-2018 | 98.18 | 96.11 | 93.21 | 92.69 |
| **With CSA** | NSL-KDD | 99.01 | 99.08 | 99.95 | 99.51 |
| | Kyoto | 98.99 | 98.93 | 98.16 | 98.53 |
| | CSE-CIC-IDS-2018 | 99.99 | 99.87 | 99.96 | 99.91 |

Tables 7–9 show a performance comparison between previous works and our proposed ensemble-based cloud intrusion detection. The studies presented in Tables 7–9 all used one of the following:

- Feature selection or feature extraction methods in addition to one of the machine learning models for classification.
- Deep learning models to determine and classify features.
- A method to handle the problem of imbalanced datasets along with ML or DL models.
- Ensemble learning model based on majority voting.
- One of the optimization algorithms in addition to ML or DL models.

Our proposal differs from others because it begins with preparing the dataset and addressing the issue of imbalance. Next, we formed a comprehensive feature set using feature selection approaches, which was then fed into ML and DL algorithms. Additionally, the results were optimized via the crow search algorithm according to the weight-based voting approach to obtain a high performance in our ensemble learning model.

**Table 7.** Comparison of the proposed model with state-of-the-art methodologies on the NSL-KDD dataset.

| No. | Reference No. | Year Published | Methodology | ACC | R | P | F |
|---|---|---|---|---|---|---|---|
| 1 | [64] | 2019 | FFDNN | 86.62 | - | - | - |
| 2 | [65] | 2019 | ICVAE-DNN | 85.97 | 77.43 | 97.39 | 86.27 |
| 3 | [66] | 2019 | 5-layers DNN | 78.50 | 78.50 | 81.00 | 76.50 |
| 4 | [67] | 2019 | WGAN-GP | 80.80 | - | - | - |
| 5 | [68] | 2019 | AE-RL | 80.16 | 80.16 | 79.74 | 79.40 |
| 6 | [69] | 2019 | CNN-1D | 78.97 | - | - | - |
| 7 | [70] | 2019 | AFSA-GA-PSO-DBN | 82.36 | - | - | - |
| 8 | [71] | 2019 | FNN-LSO | 94.04 | 89.83 | 97.43 | 93.05 |
| 9 | [72] | 2019 | MDPCA-DBN | 82.08 | 70.51 | 97.27 | 81.75 |
| 10 | [73] | 2019 | RNN-ABC | 95.62 | 95.84 | - | - |
| 11 | [74] | 2019 | GA+DBN | - | 97.67 | 97.36 | - |
| 12 | [24] | 2020 | BAT-MC | 84.25 | - | - | - |
| 13 | [29] | 2020 | CFS-BA-Voting(C4.5,RF,ForestPA) | 99.81 | 99.80 | 99.8 | 99.8 |
| 14 | [33] | 2020 | DSSTE+AlexNet | 82.84 | 82.78 | 83.94 | 81.66 |
| 15 | [40] | 2020 | AESMOTE | 82.09 | 82.09 | 84.11 | 82.43 |
| 16 | [75] | 2020 | AE | 87.00 | 82.04 | 87.85 | 81.21 |
| 17 | [76] | 2020 | FCM-SMO | 86.00 | 88.40 | 84.70 | 86.50 |
| 18 | [77] | 2020 | C5+OC-SVM | 83.24 | - | - | - |
| 19 | [78] | 2020 | Multi-CNN fusion | 81.33 | - | - | - |
| 20 | [79] | 2020 | CNN-BiLSTM | 83.58 | 84.49 | 85.82 | 85.14 |
| 21 | [80] | 2020 | DRNN | 92.18 | 94.27 | 90.23 | 92.29 |
| 22 | [81] | 2020 | GA-KELM | - | 94.01 | - | - |
| 23 | [8] | 2021 | T2FNN | - | 97.30 | 98.50 | 96.00 |
| 24 | [16] | 2021 | UEFFS | 96.06 | 97.90 | - | - |
| 25 | [18] | 2021 | MFFSEM | 84.33 | 96.43 | 74.61 | 84.13 |
| 26 | [20] | 2021 | MapReduce+GA+Random Tree | 90.45 | - | - | - |
| 27 | [21] | 2021 | CAFE-CNN | 83.43 | - | - | - |
| 28 | [36] | 2021 | PTDAE+DNN | 83.33 | 83.33 | 86.02 | 82.04 |
| 29 | [82] | 2021 | OCNN-HMLSTM | 90.67 | 95.19 | 86.71 | 91.46 |
| 30 | [83] | 2021 | I-SiamIDS | 80.00 | - | - | - |
| 31 | [30] | 2022 | IG+GR+CS-SVM | 88.15 | 90.45 | 82.87 | 83.48 |
| 32 | [84] | 2022 | ABC-BWO-CONV-LSTM | 98.67 | 100 | 97.48 | 98.73 |
| 33 | [85] | 2022 | CP-GWO-O-LSTM | 96.38 | 98.63 | 97.59 | 98.11 |
| 34 | [31] | 2023 | EGA-PSO-IRF | 98.09 | 88.53 | 96.24 | 91.87 |
| | | | **Our Proposed Ensemble** | **99.01** | **99.08** | **99.95** | **99.51** |

**Table 8.** Comparison of the proposed ensemble with conventional methodologies on the Kyoto dataset.

| No | Reference No | Year Published | Methodology | ACC | R | P | F |
|----|----|----|----|----|----|----|----|
| 1 | [86] | 2014 | CSV-ISVM | - | 90.14 | - | - |
| 2 | [87] | 2015 | OS-ELM | 96.37 | 97.95 | 95.80 | 96.86 |
| 3 | [88] | 2018 | VAE-Label | - | 75.30 | 97.50 | 85.00 |
| 4 | [89] | 2018 | BA-ELM | 97.96 | 98.75 | - | - |
| 5 | [66] | 2019 | 5-layers DNN | 88.50 | 96.40 | 91.30 | 93.80 |
| 6 | [90] | 2019 | HIDS (NBFS-OSVM-PKNN) | - | 94.75 | 56.89 | - |
| 7 | [31] | 2023 | IG+GR+CS-SVM | 96.42 | 96.23 | 90.53 | 92.96 |
| | | | **Our Proposed Ensemble** | **98.99** | **98.93** | **98.16** | **98.53** |

**Table 9.** Comparison of the proposed ensemble with recent methods on the CSE-CIC-IDS 2018 dataset.

| No | Reference No | Year Published | Methodology | ACC | R | P | F |
|----|----|----|----|----|----|----|----|
| 1 | [91] | 2019 | SMOTE-LSTM+AM | 96.20 | 96.00 | 96.00 | - |
| 2 | [92] | 2019 | CNN | 95.14 | - | - | - |
| 3 | [33] | 2020 | DSSTE+miniVGGNet | 96.99 | 96.97 | 97.46 | 97.04 |
| 4 | [58] | 2020 | RBM | 96.55 | 94.00 | - | - |
| 5 | [93] | 2020 | DNN+PSO | 95.00 | 98.20 | - | - |
| 6 | [94] | 2020 | DNN | 90.25 | 59.00 | 65.00 | - |
| 7 | [36] | 2021 | PTDAE+DNN | 95.79 | 95.79 | 95.38 | 95.11 |
| 8 | [95] | 2021 | HCRNN | 97.75 | 97.12 | 96.33 | 97.60 |
| 9 | [84] | 2022 | ABC-BWO-CONV-LSTM | 98.25 | 98.67 | 97.48 | 98.18 |
| 10 | [31] | 2023 | IG+GR+CS-SVM | 99.89 | 92.93 | 93.02 | 92.97 |
| | | | **Our Proposed Ensemble** | **99.99** | **99.87** | **99.96** | **99.91** |

From Tables 7–9, it can be inferred that the proposed ensemble model outperforms other strategies. Compared to others, the proposed work assigns more importance to feature selection. Feature selection plays a crucial role in improving performance metrics. Our approach was tested on multiple benchmark datasets during validation, unlike some other works, and this provides the benefit of including a variety of attacks. The proposed ensemble-based intrusion detection system focuses on generating a feature set and optimizing weights. The feature selection technique incorporates both filter and automated approaches to create an extensive feature set, and the usage of the ensemble model over a single classifier results in a better accuracy.

From Figure 3a, it can be inferred that the proposed ensemble model has the lowest FPR value, making it the best-performing model among those tested in terms of minimizing false positive detections. Conversely, the XGBoost model has the highest FPR value, indicating a higher likelihood of misclassifying benign network activities as attacks. The other models, SVM, LSTM, and FLN, follow in that order. Figure 3b demonstrates that the proposed ensemble model has the lowest FNR value, establishing it as the top-performing model among those evaluated for reducing false negative detections. In contrast, the XGBoost model possesses the highest FNR value, suggesting an increased probability of incorrectly categorizing attacks as normal activities. The remaining models, SVM, LSTM, and FLN, are ranked in that order according to their FNR values.

Overall, Figure 3 presents a comparison of FPRs and FNRs between each of the individual models (SVM, XGBoost, LSTM, and FLN) used in the ensemble model and our proposal tested on the NSL-KDD dataset. A performance comparison was conducted to show the strength of the ensemble model over individual models in the ensemble. The performance comparison shows that the FPR and the FNR are reduced when compared with the individual models.

Figure 4a reveals that the proposed ensemble model exhibits the lowest FPR value, while the XGBoost model presents the highest FPR value. As for the remaining models, the

FPR values decrease in the order SVM, FLN, and LSTM. Notably, the FPR values for SVM and FLN are identical. Similarly, Figure 4b explains the FNR values in the same manner.

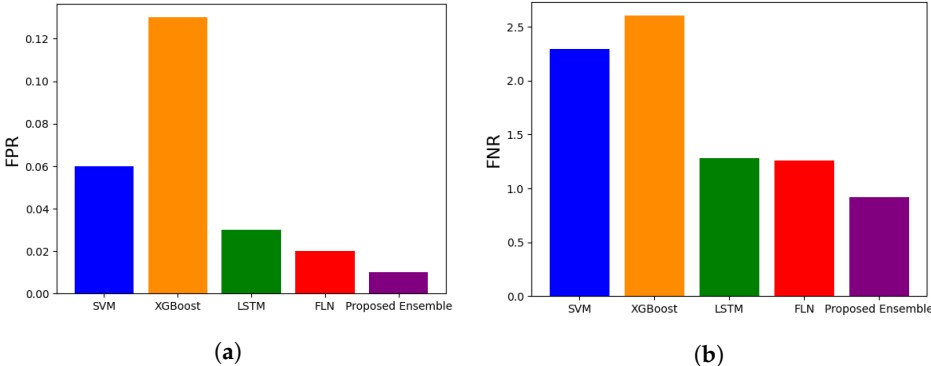

**Figure 3.** FPR and FNR comparison between the individual models and ensemble model for the NSL-KDD dataset. (**a**) FPR and (**b**) FNR.

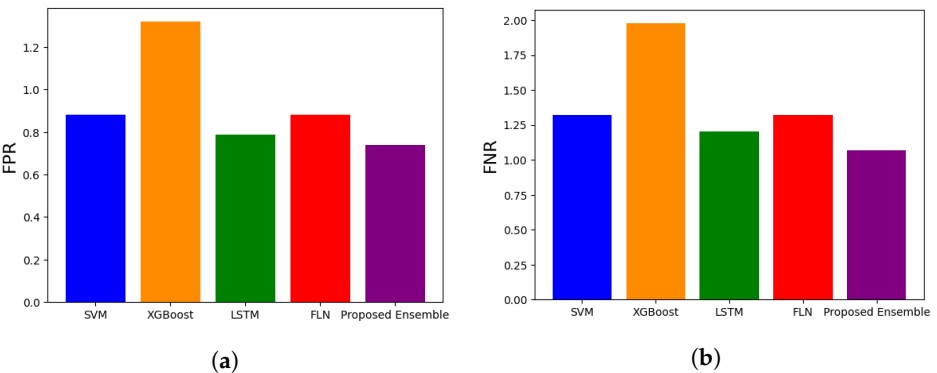

**Figure 4.** FPR and FNR comparison between the individual models and ensemble model for the Kyoto dataset. (**a**) FPR and (**b**) FNR.

The overall FPR and FNR comparisons for the Kyoto dataset are shown in Figure 4. The ensemble model's FPR and FNR are lower than the individual ML models, as can be observed.

In Figure 5, the performance of the FPR and FNR of the CSE-CIC-IDS 2018 shows a decline compared to the ML and DL models used in the ensemble. Figure 5a illustrates that the proposed ensemble model showcases the minimum FPR value, whereas the XGBoost model has the maximum FPR value. Regarding the other models, SVM, LSTM, and FLN, they are arranged according to their individual FPR values. It is worth noting that the FPR values for LSTM and FLN are the same. In a similar fashion, Figure 5b describes the FNR values.

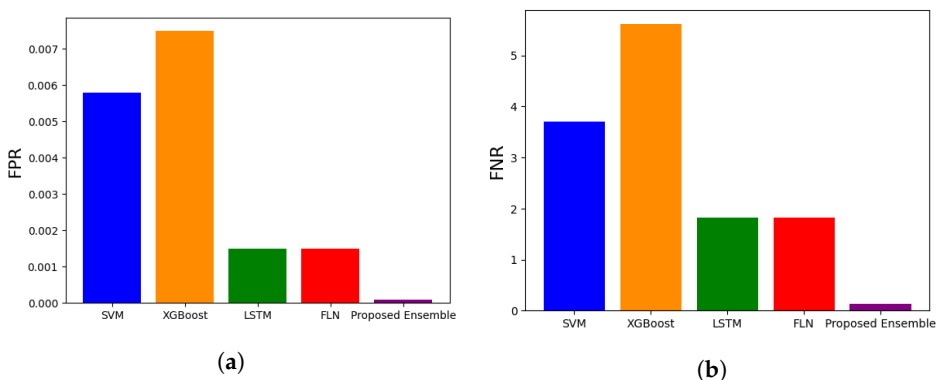

**Figure 5.** FPR and FNR comparison between individual models and ensemble model for CSE-CIC-IDS 2018. (**a**) FPR and (**b**) FNR.

Figures 3–5 show that the proposed ensemble model was able to achieve a lower FPR and FNR, indicating improved classification accuracy. The occurrence of falsely reporting data as an intrusion is reduced in all types of attacks. In summary, a lower FPR signifies that the model has a reduced chance of incorrectly identifying benign network activities as malicious. Similarly, a lower FNR indicates that the model has a decreased likelihood of misclassifying attacks as benign network activities.

Figures 6–8 illustrate the confusion matrices of our proposed ensemble. Each dataset (NSL-KDD, Kyoto, and CSE-CIC-IDS 2018) has its own confusion matrix representing the performance of our proposed ensemble. In the case of these datasets (NSL-KDD, Kyoto, and CSE-CIC-IDS 2018), the proposed ensemble is trained to predict the type of network attack or normal traffic from a total of 5, 3, and 15 categories (including normal traffic), respectively. These datasets are used for evaluating intrusion detection systems. The confusion matrix is a square matrix, where each row represents the actual class (true labels) and each column represents the predicted class (predicted labels). The diagonal elements represent correct predictions, while the off-diagonal elements represent misclassifications.

Figure 6 shows the confusion matrix of the NSL-KDD dataset. In this matrix, the first row and the first column represent the number of DoS attack instances. The proposed model has correctly predicted 7459 DoS attack instances with only one misclassification, where it predicted the normal category instead. The second row and the second column represent the number of probe attack instances. The proposed model accurately predicted 2421 instances of this attack type without any misclassifications. The same analysis can be applied to the remaining three types.

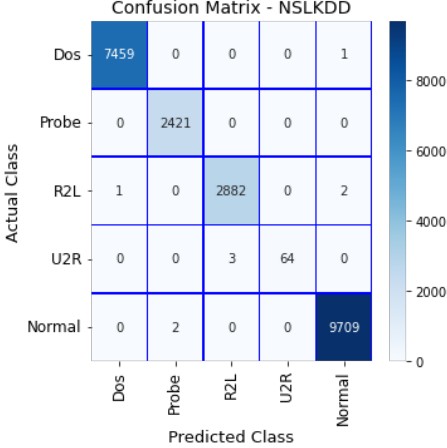

**Figure 6.** Confusion matrix of the NSL-KDD dataset.

Figure 7 displays the Kyoto dataset's confusion matrix. Within this matrix, the first row and column correspond to the count of known attack instances. The suggested system accurately identified 75,759 instances of known attacks. Nevertheless, 52 and 134 misclassifications occurred where the model predicted normal traffic and unknown attack categories, respectively. The second row and column denote the number of normal traffic instances. The proposed system successfully predicted 10,987 normal traffic instances. However, there are 263 instances where the model incorrectly identified a known attack. No cases were misclassified as the unknown attack category. This analysis can be extended to the unknown attack type as well.

Figure 8 showcases the confusion matrix for the CSE-CIC-IDS 2018 dataset. Within this matrix, the first row and column pertain to the count of benign traffic instances. The recommended system precisely identified 36,003 instances of benign traffic, with no misclassifications. The second row and column signify the number of bot attack instances. The proposed system effectively predicted 3759 bot attack instances, also without any misclassifications. This analysis can be similarly applied to the remaining thirteen types of attacks.

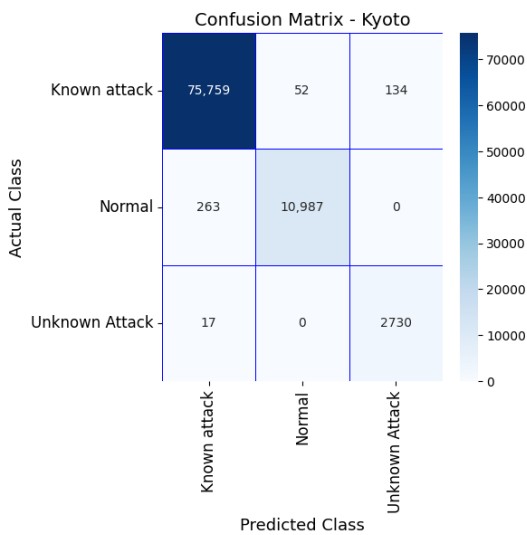

**Figure 7.** Confusion matrix of Kyoto dataset.

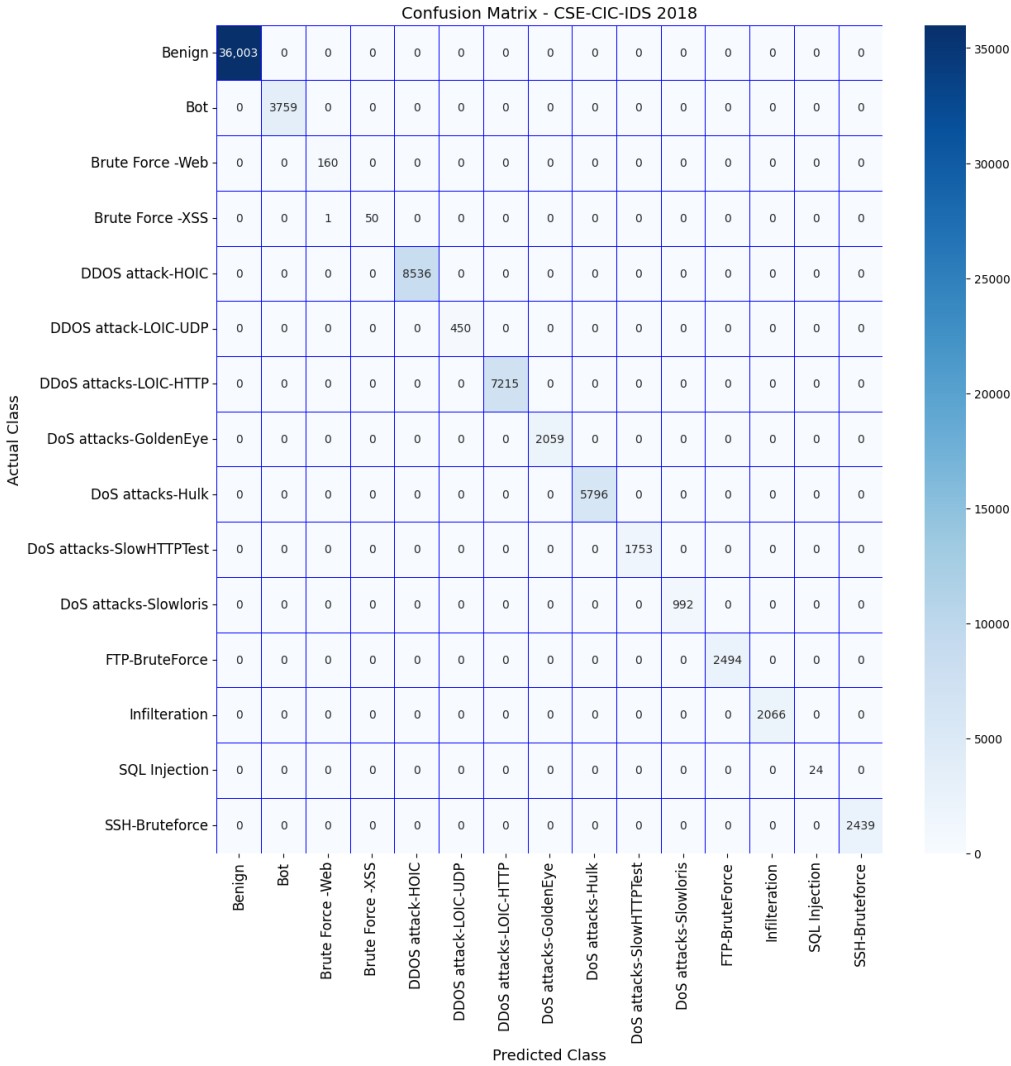

**Figure 8.** Confusion matrix of the CSE-CIC-IDS 2018 dataset.

In summary, the confusion matrices demonstrate that the proposed ensemble system boasts a high degree of accuracy in detecting various network attack types and normal traf-

fic, exhibiting outstanding performance on all three datasets. The most correct predictions lie on the diagonal, with only a minimal number of misclassifications. For a thorough assessment of the proposed system's performance, it is crucial to take into account additional metrics such as recall, FAR, FNR, precision, and F1-score, alongside the confusion matrix. These metrics are detailed in Tables 10–12.

Based on Figure 6, the performance metrics of the classes in the NSL-KDD dataset are calculated, as shown in this Table 10.

**Table 10.** Performance metrics of various classes by the proposed ensemble model for the NSL-KDD dataset.

| Attack Type | ACC | R | FAR | FNR | P | F |
|---|---|---|---|---|---|---|
| Normal | 99.98 | 99.98 | 0.02 | 0.02 | 99.97 | 99.97 |
| DOS | 99.99 | 99.99 | 0.01 | 0.01 | 99.99 | 99.99 |
| Probe | 99.99 | 100.00 | 0.01 | 0.00 | 99.92 | 99.96 |
| R2L | 99.97 | 99.90 | 0.02 | 0.10 | 99.90 | 99.90 |
| U2R | 99.99 | 95.52 | 0.00 | 4.48 | 100.00 | 97.71 |

Figure 9 shows that our approach outperformed other traditional works in terms of detection rate, false alarm rate, and accuracy for the various types of classes in the NSL-KDD dataset. Most of the attack types were detected with a higher detection rate and a reduced FAR.

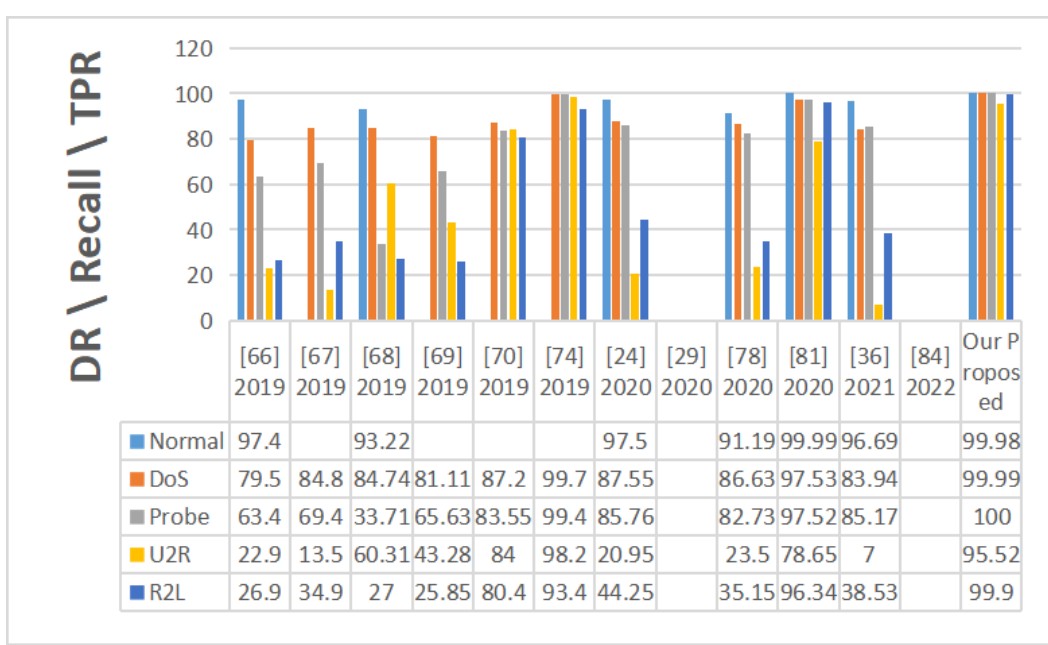

**Figure 9.** *Cont.*

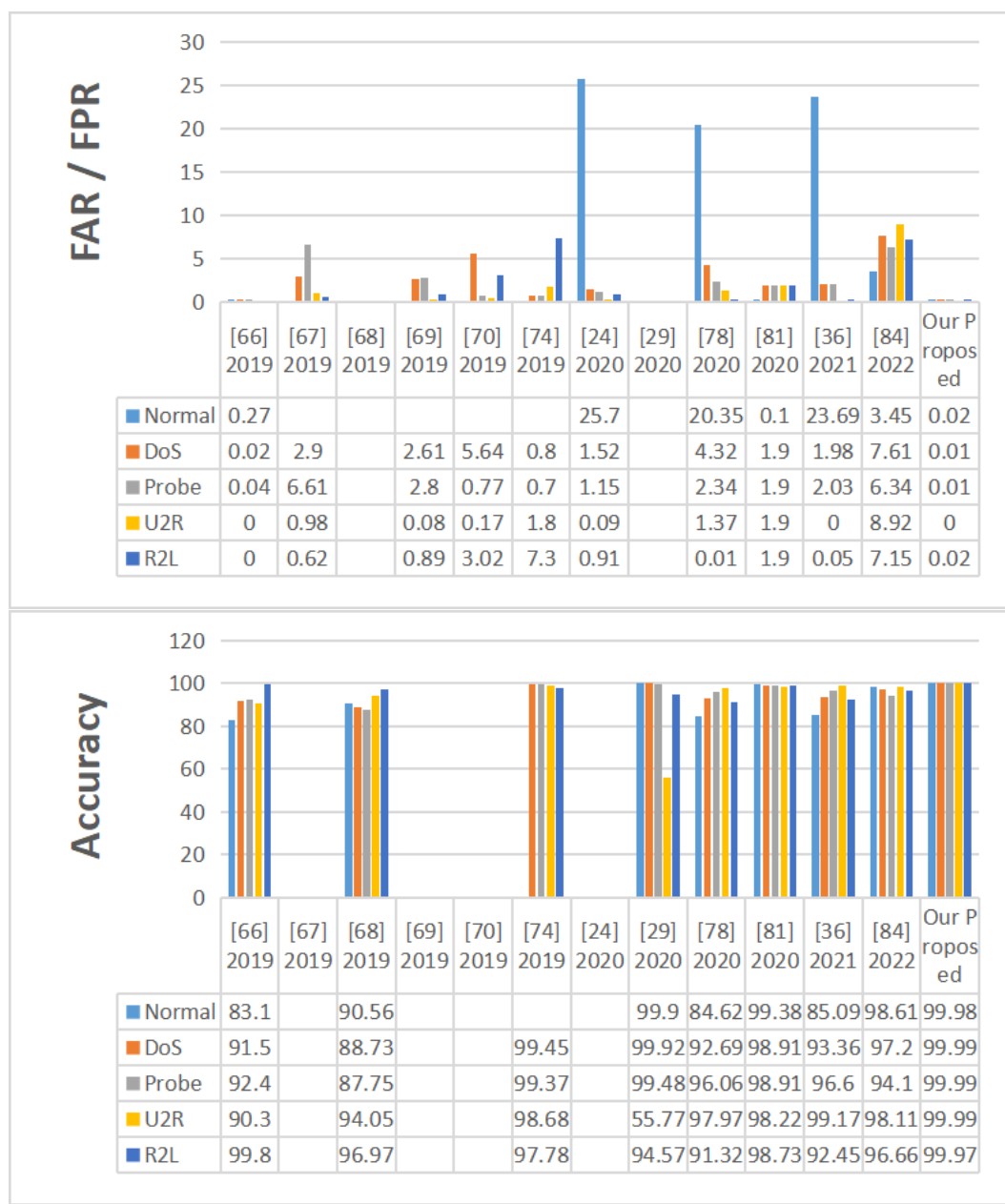

**Figure 9.** Comparison of the performance of various categories between conventional models and the proposed ensemble model for the NSL-KDD dataset [24,29,36,66–70,74,78,81,84].

The performance metrics of the classes in the Kyoto dataset were determined using Figure 7, as presented in Table 11, which shows that normal, known, and unknown attacks were detected with a high performance. To our knowledge, we did not find any previous work containing a similar analysis of performance metrics for the Kyoto dataset to compare the results of our work with.

**Table 11.** Performance metrics of types of categories by the proposed ensemble for the Kyoto dataset.

| Attack Type | ACC | R | FAR | FNR | P | F |
|---|---|---|---|---|---|---|
| Normal | 99.65 | 97.66 | 0.07 | 2.34 | 99.53 | 98.59 |
| Known Attack | 99.48 | 99.76 | 2.00 | 0.24 | 99.63 | 99.69 |
| Unknown Attack | 99.83 | 99.38 | 0.15 | 0.62 | 95.32 | 97.31 |

Figure 8 was used to generate the performance metrics for the classes in the CSE-CIC-IDS-2018 dataset, as shown in Table 12.

**Table 12.** Performance metrics of types of classes by the proposed ensemble for the CSE-CIC-IDS-2018 dataset.

| Attack Type | ACC | R | FAR | FNR | P | F |
|---|---|---|---|---|---|---|
| Benign | 100.00 | 100.00 | 0.00 | 0.00 | 100.00 | 100.00 |
| Bot | 100.00 | 100.00 | 0.00 | 0.00 | 100.00 | 100.00 |
| Brute Force-Web | 100.00 | 100.00 | 0.00 | 0.00 | 99.38 | 99.69 |
| Brute Force-XSS | 100.00 | 98.04 | 0.00 | 1.96 | 100.00 | 99.01 |
| SQL Injection | 100.00 | 100.00 | 0.00 | 0.00 | 100.00 | 100.00 |
| DDOS attack-HOIC | 100.00 | 100.00 | 0.00 | 0.00 | 100.00 | 100.00 |
| DDOS attack-LOIC-UDP | 100.00 | 100.00 | 0.00 | 0.00 | 100.00 | 100.00 |
| DDoS attacks-LOIC-HTTP | 100.00 | 100.00 | 0.00 | 0.00 | 100.00 | 100.00 |
| DoS attacks-GoldenEye | 100.00 | 100.00 | 0.00 | 0.00 | 100.00 | 100.00 |
| DoS attacks-Hulk | 100.00 | 100.00 | 0.00 | 0.00 | 100.00 | 100.00 |
| DoS attacks-SlowHTTPTest | 100.00 | 100.00 | 0.00 | 0.00 | 100.00 | 100.00 |
| DoS attacks-Slowloris | 100.00 | 100.00 | 0.00 | 0.00 | 100.00 | 100.00 |
| FTP-BruteForce | 100.00 | 100.00 | 0.00 | 0.00 | 100.00 | 100.00 |
| SSH-Bruteforce | 100.00 | 100.00 | 0.00 | 0.00 | 100.00 | 100.00 |
| Infilteration | 100.00 | 100.00 | 0.00 | 0.00 | 100.00 | 100.00 |

The following Table 13 displays that the results of our methodology have exceeded the conventional works regarding the detection rate, false alarm rate, and accuracy for the various types of classes in the CSE-CIC-IDS-2018 dataset.

**Table 13.** Comparison of the performance of various classes between the previous works and the proposed ensemble method for the CSE-CIC-IDS-2018 dataset.

| Attack Type | Our Proposed | | | [36] 2021 | | | [58] 2020 |
|---|---|---|---|---|---|---|---|
| | ACC | R | FAR | ACC | R | FAR | R |
| Benign | 100.00 | 100.00 | 0.00 | 97.48 | 99.46 | 4.48 | - |
| Bot | 100.00 | 100.00 | 0.00 | 100.00 | 99.97 | 0.00 | 96.19 |
| Brute Force-Web | 100.00 | 100.00 | 0.00 | 99.98 | 60.00 | 0.00 | 82.22 |
| Brute Force-XSS | 100.00 | 98.04 | 0.00 | 99.99 | 74.42 | 0.00 | 83.16 |
| SQL Injection | 100.00 | 100.00 | 0.00 | 100.00 | 43.75 | 0.00 | 100.00 |
| DDOS attack-HOIC | 100.00 | 100.00 | 0.00 | 100.00 | 100.00 | 0.00 | 97.54 |
| DDOS attack-LOIC-UDP | 100.00 | 100.00 | 0.00 | 99.99 | 100.00 | 0.01 | 96.15 |
| DDoS attacks-LOIC-HTTP | 100.00 | 100.00 | 0.00 | 99.98 | 99.82 | 0.00 | 96.18 |
| DoS attacks-GoldenEye | 100.00 | 100.00 | 0.00 | 100.00 | 99.93 | 0.00 | 92.01 |
| DoS attacks-Hulk | 100.00 | 100.00 | 0.00 | 100.00 | 99.99 | 0.00 | 91.32 |
| DoS attacks-SlowHTTPTest | 100.00 | 100.00 | 0.00 | 98.33 | 51.99 | 0.43 | 93.31 |
| DoS attacks-Slowloris | 100.00 | 100.00 | 0.00 | 100.00 | 100.00 | 0.00 | 97.04 |
| FTP-BruteForce | 100.00 | 100.00 | 0.00 | 98.33 | 88.16 | 1.29 | 100.00 |
| SSH-Bruteforce | 100.00 | 100.00 | 0.00 | 100.00 | 99.97 | 0.00 | 100.00 |
| Infilteration | 100.00 | 100.00 | 0.00 | 97.50 | 23.84 | 0.27 | 96.41 |

However, we did not focus on reducing the consumption of computational and temporal resources in our system; our priority was on designing a comprehensive model that overcomes some of the most important drawbacks that were presented in previous studies and that can be implemented on the most three popular datasets for detecting the largest number of attacks with an outstanding performance according to the comparison in Tables 7–9 in terms of accuracy, recall, precision, and F-measure.

### 4.4. Limitations and Constraints of the Study

The limitations and the constraints of the study are explained below:

1. Lack of a systematically collected dataset: Creating a dataset is an expensive procedure that requires a lot of money and high-level expertise. As a result, one of the key difficulties for IDS is the systematic creation of an up-to-date dataset with sufficient examples of practically all the intruder types. To aid the research community, the dataset should be regularly updated to incorporate the most recent attack records. In the current work, datasets with older (NSL-KDD and Kyoto) and newer (CSE-CIC-IDS-2018) attacks were used to test and validate the proposal. However, we emphasize the requirement for an up-to-date comprehensive dataset that represents new attacks on real environment networks that exist today. By including the definition of the largest number of intrusions in a dataset, the system will be able to discover additional patterns and offer a defense against the greatest number of zero-day attacks.

2. An unbalanced dataset has an impact on performance: It is noted from the current study that we used the SMOTE technique to increase the number of minority attack instances to balance the dataset, which led to an increase in the data size and thus the computational resources needed. Therefore, we can use random undersampling to decrease the number of instances in the majority class, then use any existing technologies such as SMOTE to oversample the minority class to balance the class distribution. However, we may try alternative approaches, such as the Difficult Set Sampling Technique (DSSTE), Adaptive Synthetic Sampling (ADASYN), RandomOverSampler (ROS), etc, to decrease the dataset's unbalance ratio for a better performance.

3. Unknown efficiency in a real-world environment: The proposal was evaluated in a lab setting based on publicly available datasets. However, it was not put to the test in a real-world setting. Therefore, it is not yet clear how it will function in real situations. Thus, to ensure that the suggested strategy works effectively for contemporary networks, it should be tested in a real-time setting after being evaluated in a lab setting.

4. Resource consumption: DL models utilize power to learn features deeply, which provides outstanding results in detecting attacks; however, it demands a lot of time, storage, and computational resources. Bio-inspired algorithms can be used in the future to improve the effectiveness and intelligence of feature selection, reduce computational resources, and achieve better results.

## 5. Conclusions and Future Work

The security of cloud computing is of utmost importance with the rise in the application of cloud computing in the majority of public and private organizations. The proposed intrusion detection system focused on designing a model that can improve intrusion detection with a higher accuracy using some advancements in the feature selection part by combining filter and automated feature selection and optimizing weights with the CSA in the ensemble model. This resulted in better results than other state-of-the-art models. The proposed approach is capable of detecting various types of attacks with a higher detection rate and a reduced false alarm rate.

In our future work, we will strive to compute the consumption of computational resources and perform a comparison with other approaches. Furthermore, bio-inspired algorithms can be implemented to optimize the value. In verification, we can use the most recent datasets that have a wide spectrum of attacks that reflect real networks, which will be prepared well to create a balanced dataset for a better performance.

**Author Contributions:** M.B. and R.R.K. contributed the main idea of qualitative analysis for the research and wrote the original manuscript. A.A.A., S.K.B., Z.A., N.P., S.K. and A.A. contributed significantly to improving the technical and grammatical contents of the manuscript. A.A.A., S.K.B., Z.A., N.P., S.K. and A.A. reviewed the manuscript and provided valuable suggestions to further refine the manuscript. All authors have read and agreed to the published version of the manuscript.

**Funding:** Researchers Supporting Project number (RSP2023R476), King Saud University, Riyadh, Saudi Arabia.

**Data Availability Statement:** The datasets are public and already cited. The data used to support the findings of this study are available from the corresponding author upon request.

**Acknowledgments:** The authors would like to acknowledge the Researchers Supporting Project number (RSP2023R476), King Saud University, Riyadh, Saudi Arabia.

**Conflicts of Interest:** The authors declare no conflict of interest.

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
