# Peer review of "Efficient Intrusion Detection System in the Cloud Using Fusion Feature Selection Approaches and an Ensemble Classifier"

_electronics, doi:10.3390/electronics12112427_

Round 1

Reviewer 1 Report

The paper discusses intrusion detection using  Fusion Feature Selection and Ensemble Classifier.  Different datasets have been used for the performance measures.  Also, different related work has been compared.  However, the following points need to be clarified: 

-       the Ensemble learning techniques are not new and have been used in many previous research papers.  So, the reviewer would like to understand the difference between your paper and previous work. For instance, the following paper is almost doing the same. What is the difference between your work and the work done in this paper?  Also, the authors compared with many previous works but not this one, as I can see. 
https://www.sciencedirect.com/science/article/abs/pii/S1389128619314203

-       The paper title includes the word "Cloud"; However, I can not see the use of the cloud in the paper.  

-       One of the authors recently published part of this work in a conference entitled " Developing a Cloud Intrusion Detection System with Filter-Based Features Selection Techniques and SVM Classifier." So, how is this paper different from the published paper, and how the produced result could differ from the work done here? 

-       Too many references where it is not a survey paper; the authors could select only the most related work. Also, how the authors got the previous work results? Have they just reviewed the papers or redeveloped the algorithms because each paper might be based on different conditions and settings?  

Those points must be considered and clarified in the paper before publication. 

It is Ok. 

Author Response

We appreciate the efforts made by the reviewer during the process of reviewing this manuscript. We pay our sincere thanks to the esteemed reviewer for the valuable comments and suggestions. The paper has now been revised according to the suggestions of the Reviewer. Please find attached our responses to the issues raised by the reviewer.

Reviewer 2 Report

The authors cover an important and timely topic. however, there is a list of improvements expected to be made before the paper can accepted. They are elaborate don below.

Introduction states "Some of the attacks faced by cloud computing are such 26 as denial-of-service (DoS), routing information, distributed denial-of-service (DDoS), SQL 27 injection attacks, cross-site scripting XSS attacks", however, the authors refer to their own article, where, if the aritcle investigated this issue specifically, this should be mentioned briefly noting the method used in such a study, BUT in any case, other references are expected to be added to support this statement and provide a form of evidence.

Several relevant and recent (2022) papers on the IDS vs ML are missing.

Otherwise, I find the paper well-written!

Language should be improved - the paper is well-written, but there tend to be some language imperfections, which could be resolved by critically rereading the paper, identifying and eliminating those issues.

E.g., "From the tables 7, 8, and 9, they are inferred that", "Our proposal is different from others because it began with...", the start of the list is not ended by semicolon but the point instead etc.

Author Response

(The authors gave the same response as above.)

Reviewer 3 Report

Authors propose a cloud intrusion detection system IDS that is focused on boosting the
classification accuracy by improving feature selection and weighing the ensemble model with the crow search algorithm (CSA).

It is a good research work, although there are some issues that should be fixed before finally accepting the paper:

- I find missing the following related work in section 2:

Conditional Variational Autoencoder for Prediction and Feature Recovery Applied to Intrusion Detection in IoT. Sensors 2017, 17, 1967.

- At the end of the related work section authors should include a paragraph explaining what other works have not done and what is going to be done in their proposal.

- In figure 1, it is not clear what does "ensemble method" means. What is done there? Is it described in section 3?, where?

- Figures 3, 4 and 5 have 2 graphs, so they should be named as a) and b).

- Authors must explain in detail each graph included in each figure.

All graphs in figures 3, 4 and 5, must have a title in their axes and the untis in the Y axis.

- Authors should eplain deeply figures 6, 7 and 8.

Author Response

(The authors gave the same response as above.)

Round 2

Reviewer 1 Report

I have no further comments on the paper.  

It is Okay, just a proof reading is required.  

Author Response

We appreciate the time spent by the reviewer during the process of reviewing this manuscript. We pay our sincere thanks to the esteemed reviewer for the valuable comments and suggestions. The paper has now been revised according to the suggestions of the Reviewer. The changes made in the revised manuscript are highlighted in BLUE color text. Kindly, find the attached file.

Reviewer 3 Report

It is a nice proposal and a good research behind it. Authors have replied my comments quite well, but the paper still needs to be improved in order to get its acceptance.

Authors should introduce better the reader to Machine Learning in security issues in the introduction section. A paper like https://doi.org/10.3390/jsan11030038 should be cited.

Figure 1 should be better explained. It is not clear why some steps are after others.  

Figure 2 is not clearly explained from the text. Moreover, layers names are not included in the figure.

I have not detected English Language mistakes

Author Response

(The authors gave the same response as above.)
